# A herpesvirus encoded Qa-1 mimic inhibits natural killer cell cytotoxicity through CD94/NKG2A receptor engagement

Xiaoli Wang[1], Sytse J Piersma[2], Christopher A Nelson[1], Ya-Nan Dai[1], Ted Christensen[1], Eric Lazear[1], Liping Yang[2], Marjolein Sluijter[3], Thorbald van Hall[3], Ted H Hansen[1], Wayne M Yokoyama[1,2], Daved H Fremont[1,4,5]*

[1]Department of Pathology and Immunology, Washington University School of Medicine, St. Louis, United States; [2]Division of Rheumatology, Department of Medicine, Washington University School of Medicine, St. Louis, United States; [3]Department of Medical Oncology, Leiden University Medical Center (LUMC), Leiden, The Netherlands; [4]Department of Biochemistry and Molecular Biophysics, Washington University School of Medicine, St. Louis, United States; [5]Department of Molecular Microbiology, Washington University School of Medicine, St. Louis, United States

**Abstract** A recurrent theme in viral immune evasion is the sabotage of MHC-I antigen presentation, which brings virus the concomitant issue of 'missing-self' recognition by NK cells that use inhibitory receptors to detect surface MHC-I proteins. Here, we report that rodent herpesvirus Peru (RHVP) encodes a Qa-1 like protein (pQa-1) via RNA splicing to counteract NK activation. While pQa-1 surface expression is stabilized by the same canonical peptides presented by murine Qa-1, pQa-1 is GPI-anchored and resistant to the activity of RHVP pK3, a ubiquitin ligase that targets MHC-I for degradation. pQa-1 tetramer staining indicates that it recognizes CD94/NKG2A receptors. Consistently, pQa-1 selectively inhibits NKG2A[+] NK cells and expression of pQa-1 can protect tumor cells from NK control in vivo. Collectively, these findings reveal an innovative NK evasion strategy wherein RHVP encodes a modified Qa-1 mimic refractory to MHC-I sabotage and capable of specifically engaging inhibitory receptors to circumvent NK activation.
DOI: https://doi.org/10.7554/eLife.38667.001

*For correspondence:
fremont@wustl.edu

## Introduction

Herpesviruses are large double-stranded DNA viruses that are associated with malignancies and can persist in the presence of an active immune system in part through their ability to avoid detection by both cytotoxic T lymphocytes (CTL) and NK cells (*Odom et al., 2012*; *Feng et al., 2013*; *Noriega et al., 2012*), two major branches of host cellular defense in adaptive and innate immunity, respectively. CTL detect foreign antigens presented by major histocompatibility complex class I (MHC-I) by engagement of their TCR with MHC-I/peptide complexes on the surface of target cells. The importance of CTL function in control of viral infection is highlighted by numerous examples of viral proteins disrupting MHC-I antigen presentation pathways (*Hansen and Bouvier, 2009*). In contrast to CTL, functional NK cells can respond and kill viral infected cells rapidly without the need for priming and expansion. NK cells are also vital in the control of herpesvirus infection. In humans, for example, individuals with selective NK cell deficiencies, either in number or function, often exhibit

recurrent and severe herpesvirus infections including α−herpesviruses (VZV, HSV), β−herpesviruses (CMV) and γ−herpesviruses (EBV) (reviewed in (*Orange, 2013*)). Similarly, NK cell-depletion confers susceptibility upon inbred strains of mice otherwise resistant to murine cytomegalovirus (MCMV) infection (*Scalzo et al., 1992*). The pivotal role of NK cells in host defense to viral infection is also underscored by the co-evolutionary development of viral NK evasion strategies and NK receptor expansions (*Carrillo-Bustamante et al., 2016*).

NK cell activity against viral-infected targets is balanced by integrated signals from inhibitory and activating receptors. In addition to presenting peptides that activate antigen-specific CTL, MHC-I molecules also serve as ligands for NK cell inhibitory receptors. The normal, ubiquitous expression of MHC-I protects healthy cells from NK killing. When MHC-I antigen presentation is sabotaged by pathogens, NK cells are released from inhibition and consequently may attack infected cells. The loss of surface MHC-I is often referred to a 'missing-self' (*Jensen et al., 2004*; *Yokoyama et al., 2010*; *Natarajan et al., 2002*). Thus, in the battle against viral infections, cell surface MHC-I plays a pivotal role in regulating CTL and NK cells and thereby both adaptive and innate immunity. This defense strategy presents a challenge for the virus; to avoid CTL detection, it must downregulate MHC-I surface expression, yet at the same time it must avoid triggering NK cell activation due to 'missing-self'.

Not surprisingly, along with their CTL evasion mechanisms, many viruses have also evolved strategies for coping with 'missing-self' attack by NK cells. One such strategy is the selective upregulation of surface expression of HLA-E in humans and Qa-1 in mice, the non-classical MHC-I proteins that serve as ligands for the NK cell inhibitory receptor CD94/NKG2A. For example, UL40 encoded by human cytomegalovirus (HCMV) selectively stabilizes surface HLA-E by providing a nonameric peptide that is loaded on to HLA-E in the ER yet independent of transporter associated with antigen processing (TAP) (*Tomasec et al., 2000*; *Ulbrecht et al., 2000*). Other viral immune evasion proteins such as HCMV US2 and US11, Kaposi's sarcoma-associated herpesvirus (KSHV) kK5, and HIV nef, induce profound downregulation of classical MHC-I but do not compromise HLA-E expression (*Orange et al., 2002*).

The CD94/NKG2 family of receptors are type-II transmembrane heterodimers that possess C-type lectin ectodomains. They are conserved in humans and mice and expressed on large percentages of NK cells (*Petrie et al., 2008*; *Hoare et al., 2008*). Each member of this family comprises an invariant CD94 polypeptide disulfide linked to either NKG2A/B, -C, -E or -F. NKG2A and -B (a spliced form of –A) are inhibitory whereas NKG2C and -E are activating NK receptors (*Kaiser et al., 2005*; *Braud et al., 1998b*; *Lazetic et al., 1996*). The functional nature of NKG2F has not been determined. An evolutionary analysis across primates revealed that both NKG2A and NKG2C are evolving under positive selection, suggesting that both genes have been actively engaged in host-pathogen conflict throughout primate evolution (*Kaiser et al., 2008*). Notably, although the inhibitory and the activating NKG2 receptors are both specific for HLA-E or Qa-1 that in normal circumstances presents peptides dominantly derived from signal sequences of other MHC class I molecules (*Bai et al., 1998*; *Llano et al., 1998*; *Braud et al., 1998a*), the binding affinity of inhibitory CD94/NKG2A receptor to HLA-E appears higher than for the activating receptors CD94/NKG2C and NKG2E (*Kaiser et al., 2005*; *Valés-Gómez et al., 1999*), and the expression of NKG2C/E normally is extremely low (*Vance et al., 1999*). Thus, engagement of peptide HLA-E/Qa-1 complex with CD94/NKG2 receptors in normal circumstances is believed to dominantly maintain self-tolerance and to sense aberrancy in MHC-I production.

Compared to β−herpesviruses, like HCMV and MCMV, the importance of NK cells in γ−herpesvirus infection is less well understood. Opportunities to address this issue have come from the identification of rodent herpesvirus Peru (RHVP) that was isolated from a lung homogenate of a pygmy rice rat (*Oligoryzomys microtis*) caught in Peru (*Loh et al., 2011*). RHVP is a member of the rhadinovirus genus of γ−herpesviruses along with KSHV and γHV68. RHVP can establish latent infection in normal B6 and 129 mice, suggesting immune evasion mechanisms are utilized by the virus. Indeed, RHVP encodes several unique ORFs that are not present in other γ−herpesviruses and appear to have immune evasion functions (*Lubman et al., 2014*; *Lubman and Fremont, 2016*). We previously reported that the R12 protein encoded by RHVP, termed pK3 because of its similarities to the K3 proteins of KSHV and γHV68, is a MARCH (membrane-associated RING-CH) family E3 ubiquitin ligase that induces rapid degradation of MHC-I by direct interaction with the heavy chain transmembrane (TM) region in the ER (*Herr et al., 2012*). In addition, pK3 secondarily induces profound

loss of the MHC-I-dedicated chaperone tapasin as well as TAP. Thus, pK3 uses a multi-pronged attack to potently downregulate surface MHC-I. This ability to dramatically reduce antigen presentation to CTL raises the question of how RHVP copes with 'missing self' recognition by NK cells.

In this paper, we report the discovery and characterization of an RHVP encoded Qa-1 mimic that can inhibit NK activation. In contrast to Qa-1, pQa-1 lacks a canonical TM region and cytoplasmic tail, and is instead GPI anchored, which confers resistance to downregulation by RHVP pK3. Similar to cellular Qa-l, pQa-1 interacts with the CD94/NKG2A inhibitory receptor. Expression of pQa-1 inhibits NK activation in vitro and protects tumor cells from NK control in vivo. In addition, we demonstrate that pK3-induced loss of surface MHC-I renders cells susceptible to syngeneic NK killing, whereas concomitant co-expression of pQa-1 reduces cell susceptibility to NK killing in a NKG2A-dependent manner. Thus, our findings demonstrate that RHVP employs compensatory mechanisms to concurrently sabotage CTL and NK-mediated immunity.

## Results

### RHVP encodes a Qa-1-like protein that requires RNA splicing for expression

Our previous findings that pK3 of RHVP potently downregulates surface MHC-I to ablate antigen presentation to CTL (*Herr et al., 2012*) raised the question for how RHVP evades 'missing self' attack by NK cells. With this in mind, we were intrigued to find that the 5' end of the RHVP genome (1383 to 236 nucleotides in reverse direction) contains a cluster of three previously annotated ORFs (R3, R2 and R1) with high similarity to MHC-I α1, α2 and α3 domains, respectively (*Loh et al., 2011*). More strikingly, two putative introns that flank consensus splicing donor/acceptor sites (*Keller and Noon, 1984*) were detected in between the putative α1 and α2, as well as the α2 and α3 coding sequences (*Figure 1A*, upper panel). To test whether the genomic sequence 1383 to 236 nucleotides could be processed by RNA splicing into one mature mRNA encoding a complete MHC-I-like protein, total RNA was prepared from RHVP-infected IFNαβγR$^{-/-}$mouse embryonic fibroblast (MEF) cells 24 hr post-infection, from which the first strand cDNA pool was then obtained by RT-PCR using oligo dT. The sequences amplified by PCR from the cDNA pool using a forward primer to the 5' end of R3 and a reverse primer to the 3' end of R1 was consistent with predicted splicing events (*Figure 1—figure supplement 1–1A* and S1B). In these splicing events, the two introns containing 88 and 82 nucleotides, respectively, were excised (*Figure 1A*, upper panel). This organization of exons and introns is analogous to mammalian MHC-I genes as exemplified by the mouse Qa-1 gene (*Figure 1A*, lower panel), suggesting that this ORF (here annotated as pQa-1) originated as a horizontal transfer from the host to the virus. Notably, the pK3 coding sequence was also concurrently amplified from the cDNA pool using primers specific to the ORF of pK3, suggesting the two proteins are co-expressed within 24 hr post-infection (*Figure 1—figure supplement 1–1C*).

### The Qa-1-like protein is associated with β2m and expressed on the cell surface as a GPI anchored protein

Sequence comparisons using the putative mature viral protein after signal peptide cleavage (residues 1–280 corresponding to the α1–3 ectodomains) yielded strong similarity to Qa-1 proteins (>56% identity) from mouse, rat, hamster and vole and slightly less strong similarity to some classical MHC-I proteins from human, monkey, panda, wild boar and harbor seal (54–52% identity) (*Supplementary file 1*). Furthermore, phylogenetic analysis revealed that this viral MHC-I-like protein shares more similarity with rat and mouse Qa-1 than human HLA-E or mouse/human classical MHC-I (*Figure 1B*). Similar to Qa-1, it contains substitutions of Ser and Leu at the Thr143 and Trp147 residues that contribute to peptide binding in the F pocket of classical MHC-I (*Figure 1—figure supplement 1–2*). We therefore designated this RHVP-encoded protein as pQa-1 and speculated that it may be a mimic of Qa-1 capable of interacting with CD94/NKG2 receptors. However, it should be noted that distinct from Qa-1, pQa-1 was not predicted to have a canonical TM region or a C-terminal cytoplasmic tail. Instead, it was predicted by GPI-SOM (*Fankhauser and Mäser, 2005*) to potentially be GPI anchored at residue 290 (*Figure 1A*, upper panel).

Because multivalent surface expression is expected to be needed for signaling via interaction with CD49/NKG2 receptors, we next examined whether pQa-1 is surface expressed.

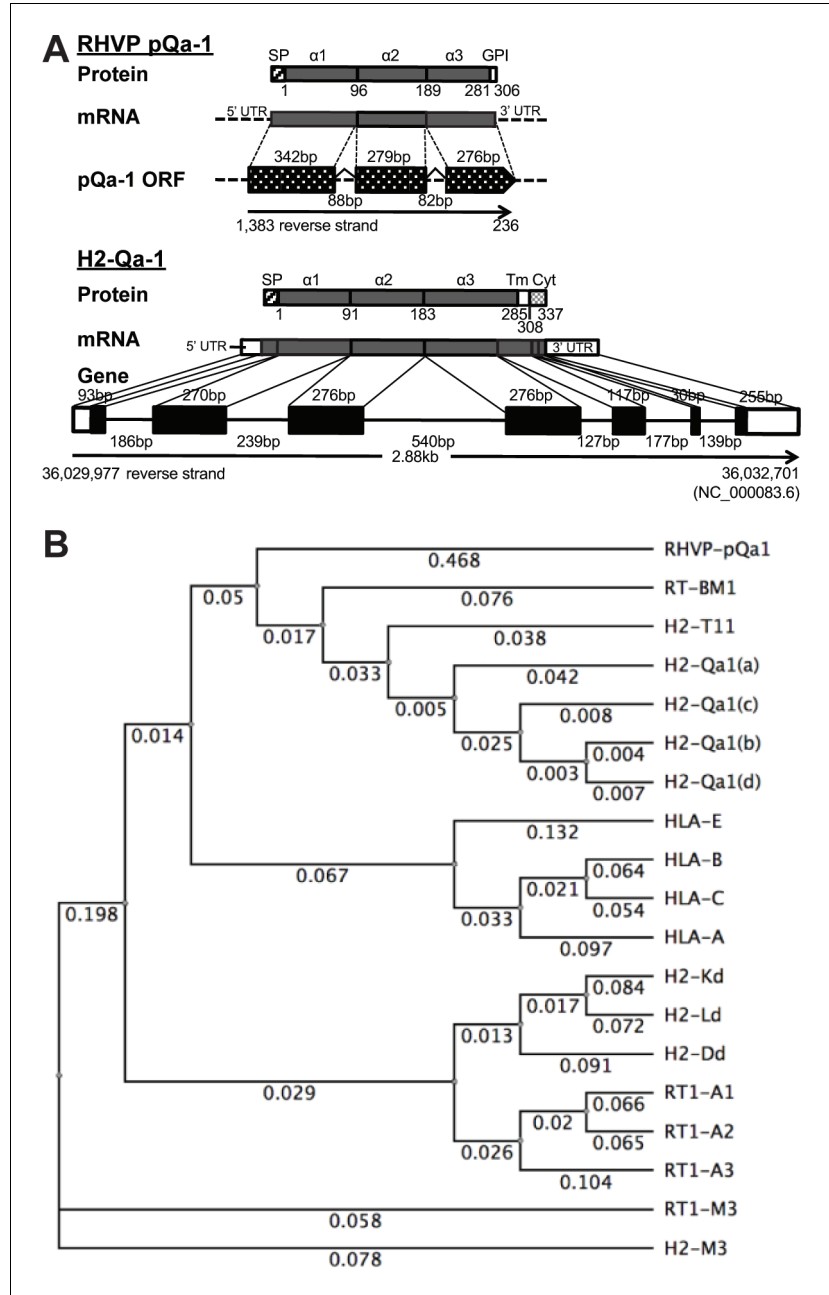

**Figure 1.** RHVP encodes a Qa-1-like protein. (**A**) Schematic comparison of the regions encoding RHVP pQa-1 and mouse H2-Qa-1. The genomic sequence is represented by a black line (solid for mouse and dotted for virus), and genome coordinates are indicated below. The exons of the translated regions of the RHVP genome and the mouse Qa-1 gene (NC_000083.6) are represented by dotted and solid black block, respectively. (**B**) Unrooted phylogenetic tree of the proteins, including RT-BM1: rat MHC-Ib-S3 (NP_001008886.2); H2-T11: mouse H2-T11 (NP_001257934); H2-Qa1(a), -Qa1(c), -Qa1(b) and -Qa1(d): mouse Qa-1a (XP_003945787), Qa-1c (AAD12244.1), Qa-1b (NP_034528) and Qa-1d (AAD31381); HLA-E: human HLA-E (NP_005507.3); HLA-B: human HLA-B7 (AAA91229); HLA-C: human HLA-Cw0702 (CAA05125); HLA-A: human HLA-A2 (ASA47534); H2-K[d]: mouse H2-K[d] (P01902.1); H2-L[d]: mouse H2-L[d] (P01897.2); H2-D[d]: mouse H2-D[d] (P01900.1); RT1-A1: rat MHC-Ia-A1 (NP_001008827.1); RT1-A2: rat MHC-Ia-A2 (NP_001008829); RT1-A3: rat MHC-Ia-A3 (NP_001008830); RT1-M3: rat MHC-Ib-M3 (NP_075210.2); H2-M3: mouse H2-M3 (AAA39597), was created using PHYLIP Neighbor Joining algorithm (http://evolution. genetics.washington.edu/phylip/phylipweb.html). Horizontal branch lengths reflect the number of nucleotide substitutions per site.

DOI: https://doi.org/10.7554/eLife.38667.002

*Figure 1 continued*

The following figure supplements are available for figure 1:

**Figure supplement 1.** Three previously annotated RHVP ORFs are spliced to form mRNA encoding an MHC-I-like protein.

DOI: https://doi.org/10.7554/eLife.38667.003

**Figure supplement 2.** Sequence alignment of pQa-1 with selected classical and non-classical MHC-I proteins.

DOI: https://doi.org/10.7554/eLife.38667.004

Since there was no specific antibody available, an HA tag was engineered into pQa-1 between the putative α3 domain and a 26 residue long C-terminal peptide predicted to contain sequences necessary for recognition by cellular GPI anchor machinery (**Figure 2A**). A retroviral IRES-GFP vector was used to stably transduce cells to express pQa-1-HA (hereafter referred to as pQa-1). Low levels of surface pQa-1 in mouse and human cells were detected by flow cytometry using an anti-HA antibody (**Figure 2B**). This low level expression of viral pQa-1 is similar to what has been reported for mouse Qa-1 (**Gays et al., 2001**). To test whether the C-terminal peptide of pQa-1 could be recognized as a GPI anchor signal, a chimeric molecule with the α1–3 of H2-L$^d$ fused to the 26 residue C-terminal peptide of pQa-1 (designated L$^d$-pQa-1) was also constructed and stably expressed on MEFs (**Figure 2A**). Surface levels of pQa-1 and L$^d$-pQa-1 but not wild type L$^d$ were both reduced upon a low dose GPI-cleaving enzyme phosphatidylinositol-specific phospholipase C (PI-PLC) treatment, indicating that both proteins were cell surface tethered via GPI anchors (**Figure 2C**, left panels). The sensitivity of L$^d$-pQa-1 to PI-PLC treatment was dose dependent and the extent of reduction in fluorescence signal upon treatment with different doses of PI-PLC is similar to that of a known GPI-anchored protein, Thy1.1, expressed on the same cells (**Figure 2C**, right panels). Further, we found that pQa-1 was associated with β2-microblobulin (β2m) by labeling pQa-1-expressing cells with [$^{35}$S] Cys/Met followed by immunoprecipitation of pQa-1, or by immunoprecipitation of pQa-1 followed by western blot of β2m (**Figure 2D**, left and right panel, respectively). In addition, we found that without β2m in a *H2-K$^{b-/-}$*, *H2-D$^{b-/-}$* and *B2m$^{-/-}$* MEF line (designated 3KO) pQa-1 was barely detected, while reconstitution of β2m into 3KO cells by retroviral transduction clearly enhanced surface expression of pQa-1 (**Figure 2E**), indicating that endogenous β2m is required for surface expression of pQa-1.

## Qdm or Qdm-like peptides stabilize pQa-1 in a manner similar to mouse Qa-1

One of the hallmarks of Qa-1 is its propensity to preferentially bind the peptide AMAPRTLLL (also called Qa-1 determinant modifier or Qdm) derived from the leader sequence of H2-D or H2-L in a TAP-dependent manner (**DeCloux et al., 1997**; **Bai et al., 1998**). NK cytotoxicity can be inhibited by the engagement of CD94/NKG2A with its ligand Qa-1 and this inhibitory effect is highly dependent on Qa-1 being loaded with the Qdm peptide (**Kraft et al., 2000**). Based on this and sequence similarity of pQa-1 to Qa-1, we tested whether pQa-1 also binds Qdm or Qdm-like peptides. To assess binding, we performed surface stabilization assays using pQa-1-expressing cells cultured with either Qdm or a control H2-K$^b$ binding peptide (ovalbumin residues 257–264, OVA). A modest but significant enhancement in pQa-1 levels was observed when co-cultured with Qdm but not the OVA peptide at 37°C for 4 hr, although the OVA peptide enhanced the level of surface K$^b$ on the same cells as reported (**Figure 3A**).

RHVP was first isolated from a pygmy rice rat trapped in Peru (**Loh et al., 2011**). Since the distribution of additional hosts is unknown, we attempted to identify a binding peptide for pQa-1 from a group of Qa-1-like candidates found in the MHC-I of different species or in some viral proteins (**Supplementary file 2**). Surface levels of pQa-1 were significantly enhanced when cells were cultured at 27°C for 4 hr with Qdm or with the Qdm-like nonamers that have one residue difference at the P1, P2 or P3 position compared to no peptide added (**Figure 3B**). In contrast to the nonamers, an octameric peptide that lacks the P1 residue of Qdm did not increase surface pQa-1. This analysis suggests that pQa-1 prefers to bind a Qdm-like nonamer and tolerates certain variations in peptide sequence. This is very similar to the observations made with mouse Qa-1 where single substitution with G/A at P1 and P3 or K at P1 in Qdm did not appear to impact peptide binding (**Kurepa and**

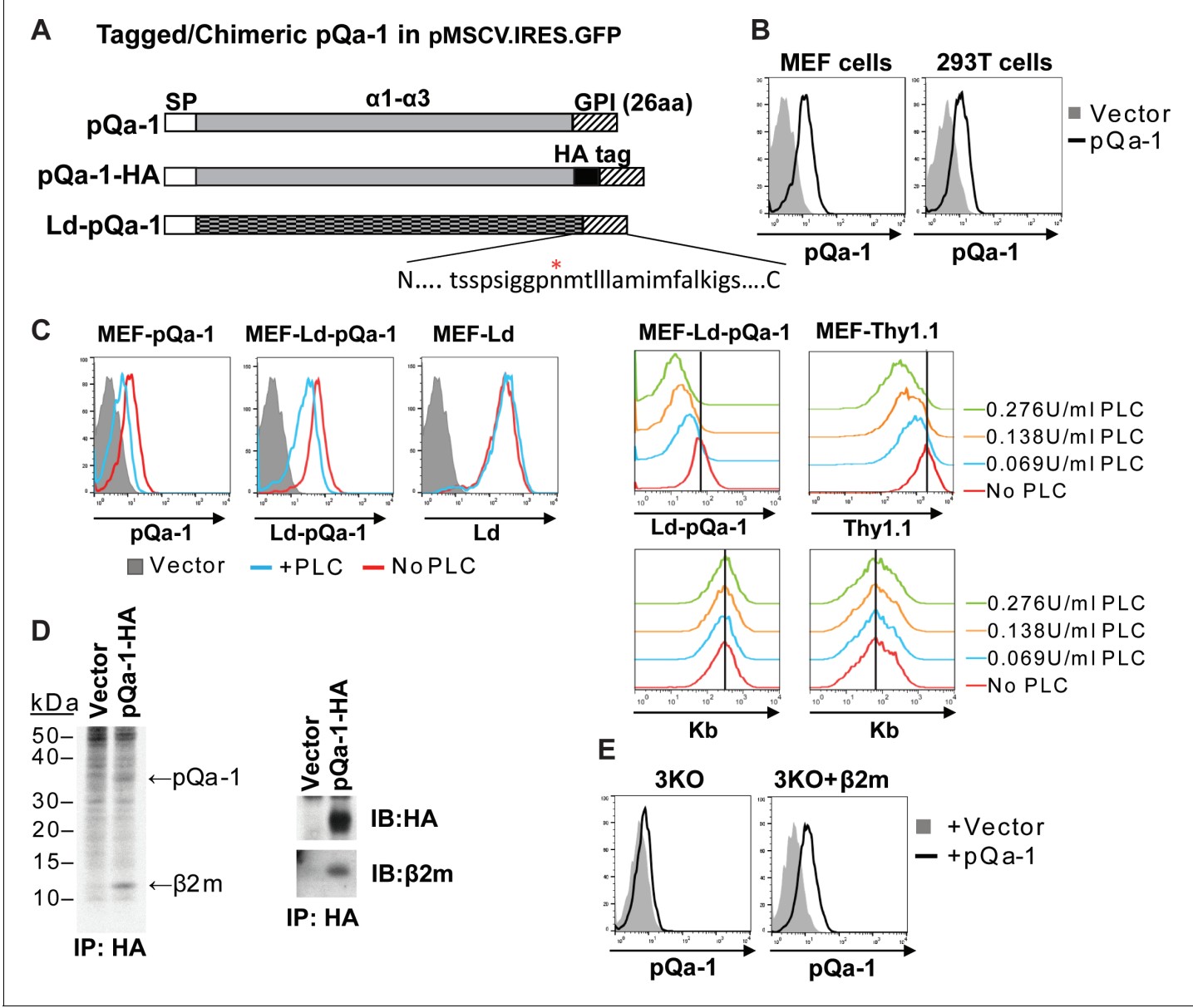

**Figure 2.** RHVP pQa-1 is GPI anchored, cell surface expressed and assembles with β2m. (**A**) Schematic depiction of the pQa-1 expression constructs used in the study. The C-terminal 26aa containing predicted GPI attachment site (marked by red star) is shown under the C-terminus of the last construct. (**B**) Mouse embryonic fibroblast (MEF) and human 293 T cells were stably transduced with the vector only or pQa-1-HA construct depicted in (**A**). Surface expression of pQa-1 on these cells was analyzed by flow cytometry using anti-HA antibody. (**C**) Left panel: cells were treated with (blue) or without (red) 0.069 U/ml phosphatidylinositol-specific phospholipase C (PI-PLC) at 37°C for 45 min before staining with anti-HA or anti-L$^d$ (30-5-7). MEFs expressing vector only served as background staining (solid gray). The representative of two independent experiments is shown. Right panel: following incubation with indicated concentration of PI-PLC, MEF cells expressing L$^d$-pQa-1 or Thy1.1 were examined. Here endogenous MHC-I (H2–K$^b$) serves as a negative control protein; its level of surface expression was unaffected by PI-PLC. (**D**) Following a 30-min pulse with $^{35}$S-Cys/Met, pQa-1 transduced MEF cells were lysed with 1% NP-40 and immunoprecipitated for pQa-1 using anti-HA. The precipitated proteins were resolved on SDS-PAGE and visualized by autoradiography (left) or immunoblotted with the indicated antibodies (right). The representative of two independent experiments is shown. (**E**) MHC-Ia- and β2m-deficient MEFs (*H2-K$^{b-/-}$ H2-D$^{b-/-}$ B2m$^{-/-}$*; 3KO) or 3KO+β2m cells transduced with pQa-1-HA or vector control were examined for surface pQa-1 expression using anti-HA. The representative of two analyses is shown.
DOI: https://doi.org/10.7554/eLife.38667.005

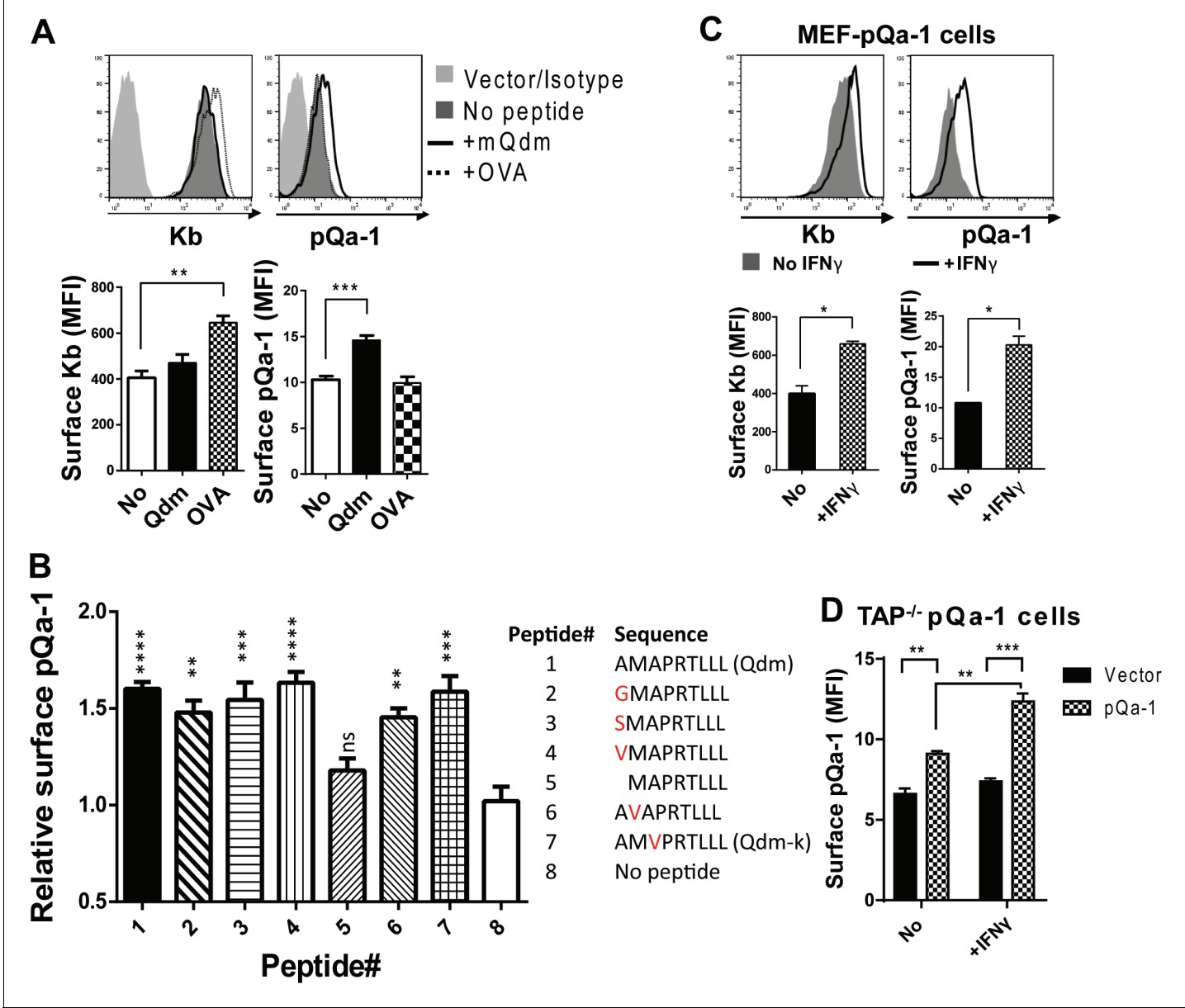

**Figure 3.** Qdm and Qdm-like peptides stabilize pQa-1 at the cell surface. (**A**) Following incubation of MEF-pQa-1cells with 100 μM Qdm (AMAPRTLLL) or OVA$_{257-264}$ (SIINFEKL) peptide at 37°C for 4 hr, surface pQa-1 and H2-K$^b$ were analyzed by flow cytometry for comparison. (**B**) MEF-pQa-1were incubated with the indicated peptide (100 μM) at 27°C for 4 hr before staining for surface pQa-1. (**C**) Surface pQa-1 was FACS analyzed after incubation with 100 U/ml IFNγ for 24 hr. (**D**) Surface pQa-1 in a stably transduced TAP1-deficient (*Tap1$^{-/-}$*) fibroblast line was detected by FACS after incubation with or without 100 U/ml IFNγ for 40 hr. Representative of two analyses with mean ±SD of duplicates is shown. Bars in the figures represent mean ±SEM of two (**C**) or three (**A and B**) independent analyses using unpaired t test or one-way ANOVA, Dunnett's multiple comparisons test to compare with no peptide controls (*=p < 0.05; **=p < 0.01; ***=p < 0.001; ****=p < 0.0001; ns = not significant). Endogenous MHC-I K$^b$ in these cells served as positive control.

DOI: https://doi.org/10.7554/eLife.38667.006

*Forman, 1997*; *Kraft et al., 2000*). These data indicate that pQa-1 maintains similar peptide-binding properties as Qa-1.

It is noteworthy that surface pQa-1 in stably transduced cells increased after IFNγ treatment (*Figure 3C*). Since expression of transduced pQa-1 was not controlled by an MHC-I promoter, this enhancement could not be explained by IFNγ promoted MHC-I production. Instead, it may be an indirect consequence of IFNγ-induced increase in β2m or/and source of stabilizing peptides. We also

examined whether pQa-1 is expressed on the surface of TAP-deficient cells. We found, similar to what we had observed with wildtype MEF cells, that TAP1-deficient fibroblasts stably transduced with pQa-1 express a low level on the cell surface, and this level can be enhanced by IFNγ treatment (*Figure 3D*). Given the role peptide loading is known to play in MHC-I quality control, we would not expect pQa-1 to be efficiently released from the ER to the surface or to survive long at the cell surface without first being loaded with peptide. Thus, expression of pQa-1 on TAP-deficient cells suggests pQa-1 can load peptide independently of TAP as has been seen for Qa-1 and HLA-E (*Lampen et al., 2013*; *Oliveira et al., 2010*). Thus, during RHVP infection, surface pQa-1 levels could be correlated with the infection-induced IFNγ production even when TAP function is impaired.

## pQa-1 is resistant to RHVP pK3-induced degradation

We previously demonstrated that pK3, a MARCH family E3 ligase encoded by RHVP, efficiently downregulates surface MHC-I by targeting newly synthesized MHC-I in the ER for rapid, ubiquitination-mediated degradation (so called ER-associated degradation, ERAD) (*Herr et al., 2012*). In contrast to mK3 of γHV68 that directly interacts with TAP to capture TAP-associated MHC-I, pK3 targets MHC-I through direct interaction with the TM region of MHC-I proteins. Newly synthesized pQa-1 includes a hydrophobic tail that is thought to be transiently embedded in the membrane before cleavage and GPI-linkage to the newly generated C terminus (*Udenfriend and Kodukula, 1995*). While the initial hydrophobic tail of pQa-1 does not have any appreciable sequence similarity to either Qa-1 or MHC-I proteins, we nevertheless sought to examine whether it might be resistant to pK3-mediated degradation. To test this, MEF cells expressing Qa-1 or pQa-1 were co-transduced with either wildtype pK3 (pK3 WT) or a pK3 RING mutant (pK3 RM) that loses ligase activity while retaining the ability to associate with the target. Indeed, in the presence of pK3 WT, Qa-1 was significantly down regulated, like classical MHC-I, even though Qa-1 was not as well expressed as pQa-1 in these experiments (*Figure 4A*, lower panel). Downregulation of Qa-1 by pK3 was more evident with higher baseline expression of endogenous Qa-1 in RMA cells (*Figure 4B*). In contrast, surface pQa-1 remained unchanged in the presence of pK3 (*Figure 4A*, upper panel). Importantly, downregulation of MHC-I and Qa-1 was not observed in the presence of the catalytically inactive pK3 RM, demonstrating that E3 ubiquitin ligase activity was required. In addition, the GPI anchored classical MHC-I chimera, $L^d$-pQa-1, was resistant to pK3, whereas the reciprocal chimera with α1-α2 domains of pQa-1 and α3-TM-cytoplasmic tail of $L^d$ (pQa-1-Ld) remained as sensitive to pK3 as wild type $L^d$ (*Figure 4C*), supporting the conclusion that pQa-1 resistance to pK3 is associated with its unique C-terminal tail. Based on these observations we speculate that pQa-1 evolved to counteract NK activation caused by the MHC-I downregulation by pK3.

## The CD94/NKG2A receptor specifically detects pQa-1/Qdm complexes

The sequence similarity and peptide-binding similarities of pQa-1 with Qa-1 suggest that it may be a ligand for CD94/NKG2 receptors on NK cells. Intriguingly, the CD94/NKG2 family consists of the inhibitory receptor CD94/NKG2A as well as activating receptors CD94/NKG2C and CD94/NKG2E. All these heterodimeric receptors have the same ligand, Qa-1 (HLA-E in humans). Antibodies that are capable of distinguishing different forms of NKG2 receptors are currently unavailable, thus the relative distribution and expression of the inhibitory versus activating NKG2 receptors remains poorly characterized. Nevertheless under normal circumstances expression of NKG2C or NKG2E at the mRNA level is much lower than NKG2A (*Vance et al., 1999*) and in NKG2A$^{-/-}$ mice, surface NKG2C is undetectable (*Rapaport et al., 2015*). With this background, we first tested whether recombinant pQa-1 tetramers detect CD94/NKG2 positive cells. To circumvent difficulties encountered in production of soluble pQa-1 proteins due to instability of peptide/pQa-1/β2m complexes, we expressed Qdm-β2m-pQa-1 as a single chain trimer (pQa-1-SCT) either GPI anchored or soluble with a biotin label at the C terminus (depicted in *Figure 5—figure supplement 5–1A*). To secure the Qdm peptide in the binding groove, we also engineered a disulfide trap between the first linker G2C and residue Y84C of the pQa-1 (pQa-1-dtSCT). As we previously observed with classical MHC-I (*Mitaksov et al., 2007*; *Truscott et al., 2007*), the pQa-1-SCT appeared to be more stable than pQa-1. The surface levels of GPI anchored pQa-1-SCT and pQa-1-dtSCT were 6–8 folds higher than pQa-1 expressed alone (*Figure 5—figure supplement 5–1A*); and their thermostabilities as measured by circular dichroism melting analysis were comparable to those reported for classical MHC-I

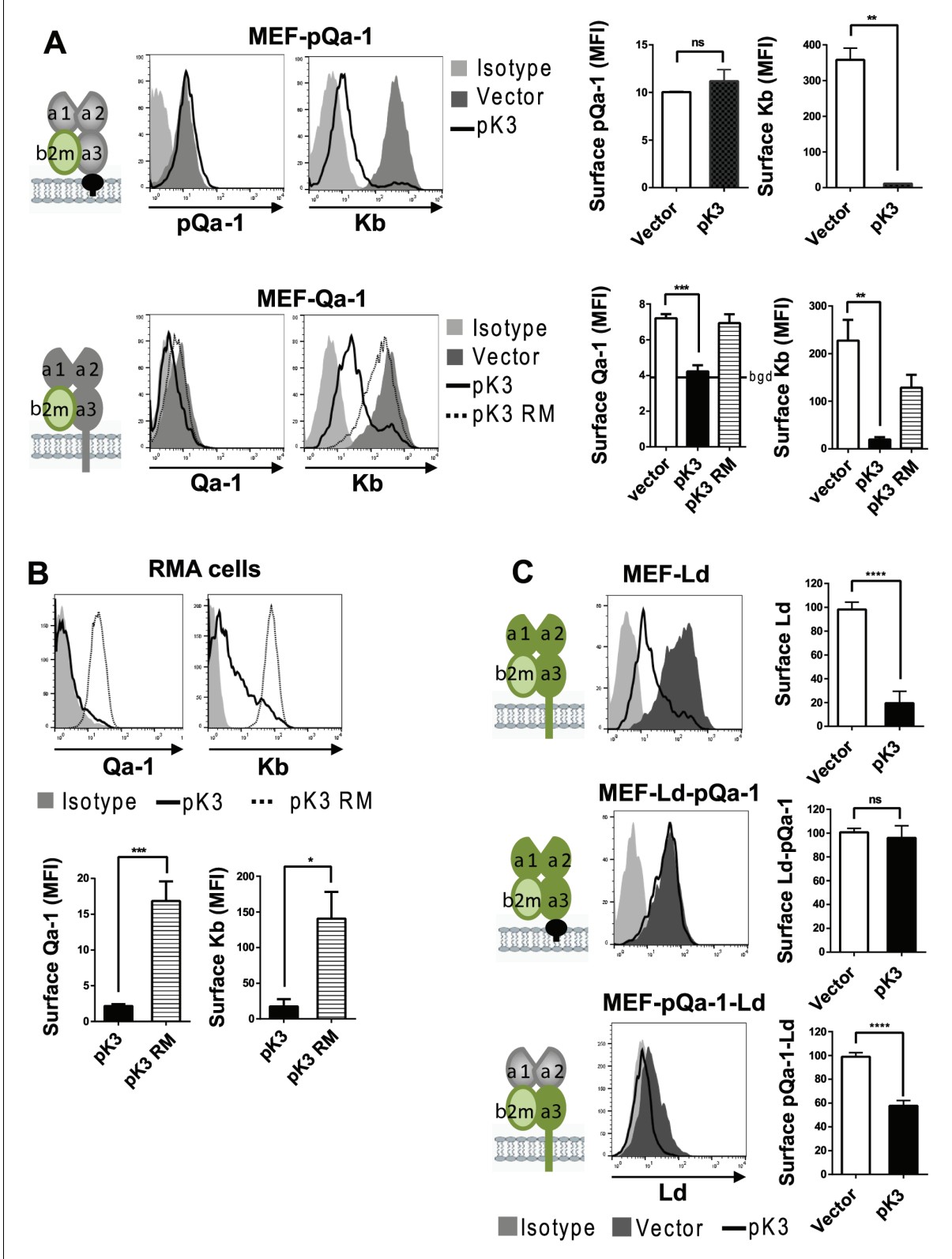

**Figure 4.** Qa-1 is sensitive to pK3-induced downregulation while pQa-1 is resistant. (A) MEF-pQa-1 (upper panel) or MEF-Qa-1 (lower panel) cells were transduced with vector, pK3 or pK3 RING mutant (pK3 RM). Surface pQa-1/Qa-1 levels were determined by anti-HA/anti-Qa-1 staining. Surface expression of endogenous MHC-I $K^b$ on these cells served as a positive control of pK3 function. (B) Surface expression of endogenous Qa-1 on RMA cells in the presence of vector, pK3 or pK3 RM were analyzed similarly. (C) Surface levels of classical MHC-I $L^d$, chimeric $L^d$-pQa-1 or chimeric pQa-1-$L^d$

*Figure 4 continued on next page*

*Figure 4 continued*

(as depicted by cartoons) on MER cells in the presence of vector or pK3 were analyzed. Quantification of median fluorescence intensity (MFI) in (**A and B**) or relative surface levels in (**C**) from two to four analyses are shown as bars with mean ±SEM. Unpaired t test or one-way ANOVA Dunnett's multiple comparisons test was used (*=p < 0.05; **=p < 0.01; ***=p < 0.001; ****=p < 0.0001; ns = not significant).
DOI: https://doi.org/10.7554/eLife.38667.007

$K^b$-SCT (*Figure 5—figure supplement 5–1B* and *Mitaksov et al., 2007*). Using soluble pQa-1-dtSCT, we next generated tetramers to stain mouse spleen cells. These pQa-1-dtSCT tetramers stained the same population recognized by the NKG2A/C/E antibody (*Figure 5*). This population consisted of about 40% of NK cells and 2% of $CD8^+$ T cells in the splenocytes isolated from wildtype C57BL/6 mice. Also similar to the level of anti-NKG2A/C/E antibody staining, the median fluorescence intensity (MFI) of $tetramer^+$ $CD8^+$ cells was lower than NK cells. Furthermore, in contrast to the wildtype mice, NK and $CD8^+$ cells from $NKG2A^{-/-}$ mice were not stained by pQa-1-dtSCT tetramer or the NKG2A/C/E antibody (*Figure 5*). Thus, pQa-1 is able to selectively engage CD94/NKG2 receptors on murine cytotoxic cells.

To further determine whether pQa-1 differentially interacts with inhibitory and activating NKG2 receptors, we evaluated cells co-expressing CD94 and either the NKG2A or NKG2C ectodomain with the TM and cytoplasmic regions of NKG2C (*Figure 5—figure supplement 5–2A*). Since expression of CD94 alone indicated by appearance of GFP was not detected by CD94 antibody in flow cytometry (*Figure 5—figure supplement 2B*), we used CD94 staining to monitor the level of the heterodimeric receptors on the cells. Using this system, we found that both the NKG2A/C/E antibody and pQa-1-dtSCT tetramer strongly stained the cells expressing CD94/NKG2A, but both reagents poorly stained cells expressing CD94/NKG2C. Nevertheless, reasonable levels of CD94 were detected in both lines (*Figure 5—figure supplement 2C*). While it is possible that pQa-1-dtSCT preferentially binds the inhibitory CD94/NKG2A rather than activating CD94/NKG2C receptor, the low level of NKG2A/C/E antibody staining of CD94/NKG2C expressing cells does not support this conclusion.

## pQa-1 expression inhibits NK cells by engagement of the CD94/NKG2A receptor

We next examined the effect of pQa-1 surface expression on NK activation ex vivo. Murine C57BL/6 splenocytes were isolated and co-cultured with CHO cells expressing vector only (CHO-V), Qa-1 or pQa-1, either wildtype (CHO-pQa-1) or pQa-1-SCT (CHO-pQa-1-SCT) (*Figure 6—figure supplement 6–1A*). CHO cells are prototypic targets of C57BL/6 NK cells and can stimulate NK cells to release cytotoxic granules and cytokines through the Ly49D activation receptor (*Idris et al., 1999*). Of note, lysosomal-associated membrane protein-1 (LAMP-1, CD107a) is a membrane component of cytotoxic granules that was recently described as a functional marker for NK degranulation, a critical step in target cell lysis (*Alter et al., 2004*). Thus, to distinguish activity of $NKG2A^+$ NK cells from $NKG2A^-$ NK cells, IFNγ and CD107a expression was analyzed using flow cytometry in the two NK populations after incubation with the indicated CHO cells.

A dose-dependent effect was observed in both types of NK cells upon incubation with increasing number of CHO cells, but the ratio of NK activation of $NKG2A^+$ to $NKG2A^-$ was stable with different amounts of offered CHO cells. A one to one ratio was therefore used to analyze the effect of Qa-1 or pQa-1 on $NKG2A^+$ NK activation. When cultured with CHO-pQa-1 cells in the presence of Qdm or Qdm-k peptide, but not a peptide that cannot stabilize surface pQa-1, mouse spleen $NKG2A^+$ NK cells showed less IFNγ production than $NKG2A^-$ cells, which was not the case upon culture with control CHO-V cells (*Figure 6A and B*). A similar decrease in IFNγ production was observed when NK cells were incubated with CHO cells expressing the natural NKG2A ligand Qa-1 (*Figure 6A and B*), suggesting that an inhibitory effect comparable to mouse Qa-1 on NK activity was mediated by the expression of viral pQa-1. In addition, like Qa-1, this suppressive effect could be reversed by the NKG2A/C/E blocking antibodies but not by an isotype control, demonstrating that pQa-1 exerts its effect specifically through the NKG2A receptor (*Figure 6C*). Furthermore, compared with the inhibitory effect mediated by CHO-pQa-1 or CHO-Qa-1 cells, CHO cells expressing pQa-1-SCT showed significantly stronger suppression on $NKG2A^+$ NK cells in IFNγ production as well as CD107a

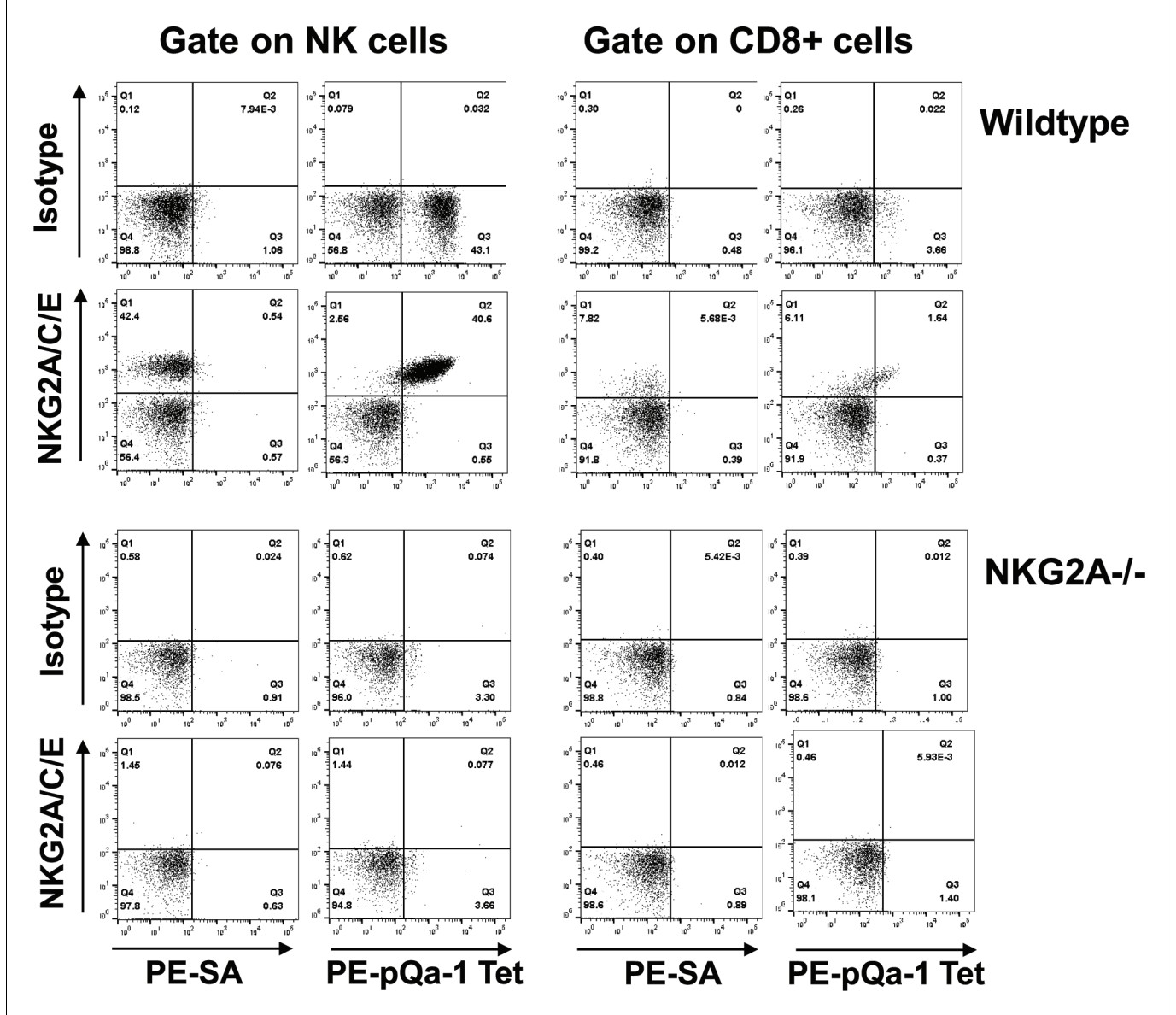

**Figure 5.** The Qdm-β2m-pQa-1 single chain trimer (SCT) specifically recognizes the CD94/NKG2A receptor. (**A**) Spleen lymphocytes isolated from wildtype C57BL/6 or NKG2A$^{-/-}$ knockout mice were stained with PE-labeled pQa-1-dtSCT tetramer (PE-pQa-1 Tet), or PE-labeled streptavidin (PE-SA) as negative control, at room temperature for 1 hr followed by staining with a mixture of fluorochrome labeled antibodies containing either anti-NKG2A/C/E (20D5) or isotype control for 30 min. The stained cells were acquired by BD Canto II and data were analyzed with FlowJo software. Representative data of four (wildtype) and two (NKG2A$^{-/-}$) independent experiments are shown. NK cells were defined as the live NK1.1$^{+}$CD3$^{-}$CD19$^{-}$ population, while CD8 T cells were gated on the live CD19$^{-}$ CD3$^{+}$CD8$^{+}$ population.

DOI: https://doi.org/10.7554/eLife.38667.008

The following figure supplements are available for figure 5:

**Figure supplement 1.** pQa-1 single chain trimer design and increased thermal stability.
DOI: https://doi.org/10.7554/eLife.38667.009

**Figure supplement 2.** pQa-1-dtSCT tetramer stains CD94/NKG2A expressing cells stronger than CD94/NKG2C-expressing cells.
DOI: https://doi.org/10.7554/eLife.38667.010

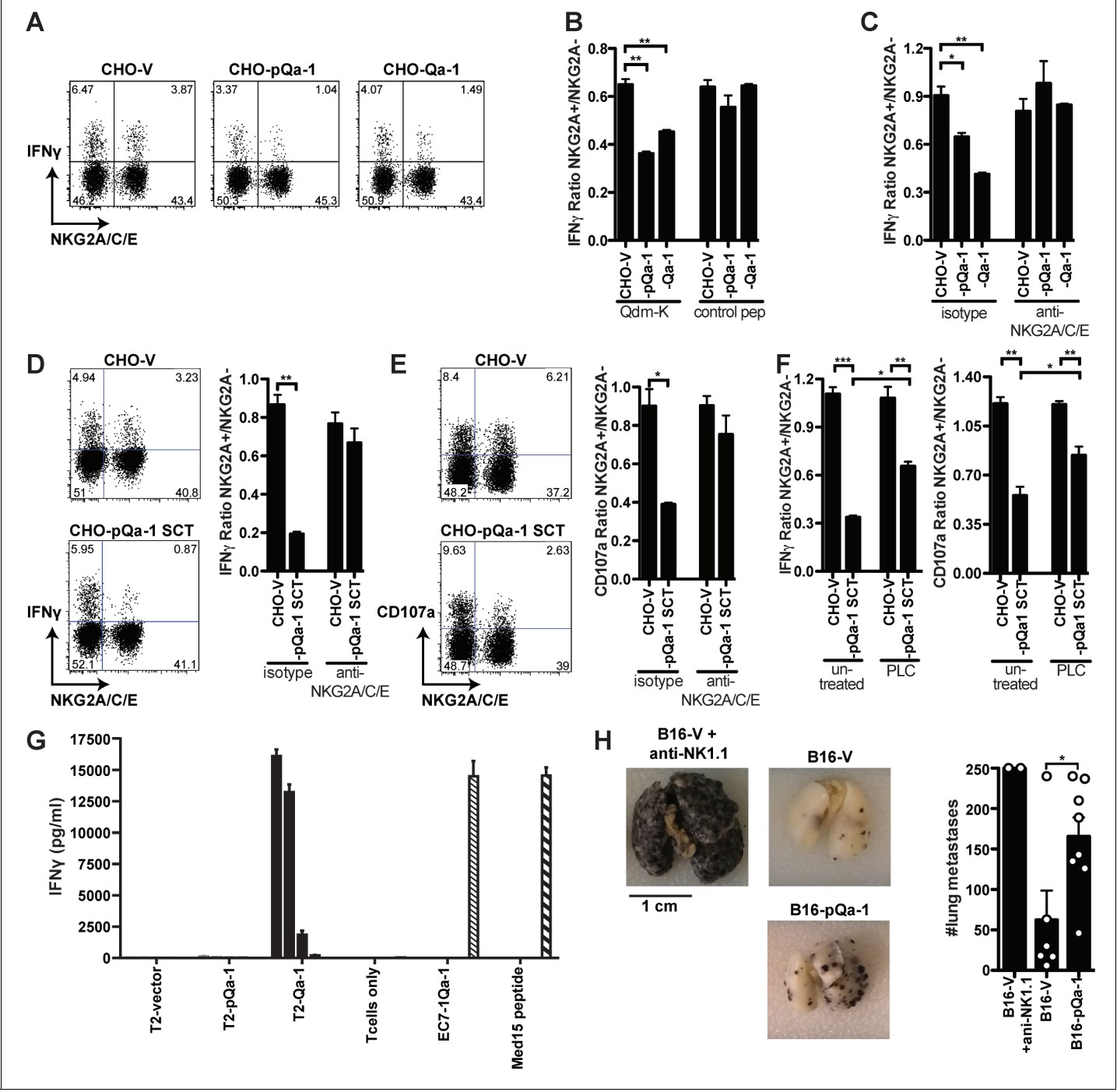

**Figure 6.** Expression of pQa-1 bound with Qdm/Qdm-like peptide inhibits NKG2A[+] NK cell activation and prevents tumor rejection in vivo. C57BL/6 splenocytes were co-cultured with CHO cells expressing the indicated constructs and the NK cells were subsequently analyzed by flow cytometry (**A–F**). The ratio of IFNγ production between NKG2A[+] and NKG2A[-] NK cells is shown in the bar graphs. (**A**) Representative dot plots of NK cell IFNγ production upon stimulation with CHO cells expressing vector, pQa-1, or Qa-1 in the presence of Qdm-k peptide (AMVPRTLLL). (**B**) Splenocytes were co-cultured with indicated CHO cells in the presence of Qdm-k or control peptide. (**C**) Co-cultures were performed as in (**B**) in the presence of isotype or 20D5 (anti-NKG2A/C/E) antibody. (**D**) IFNγ and (**E**) CD107a expression by NK cells in response to CHO cells expressing pQa-1 single chain trimer (pQa-1-SCT) was performed as in (**C**). (**F**) Same experiment as in (**D**) and (**E**) was conducted after CHO-pQa-1-SCT and CHO-V cells were treated with or without 0.5 U/ml PI-PLC. (**A–F**) Representative experiments are shown from two to three independent experiments per panel. Bars in the figures represent mean ±SEM of duplicates. (**G**) 5000 Qa-1-restricted Ln12 T cells were co-cultured overnight with titrating amounts of human T2 cells (TAP-deficient cells) expressing vector, Qa-1, or pQa-1. The amount of IFNγ in the supernatants was determined by ELISA. EC7.1-Qa-1, a mouse TAP- and

*Figure 6 continued on next page*

*Figure 6 continued*

MHC-Ia-deficient lymphoma cell line transduced to express Qa-1 served as a positive control. Mean ±SD of triplicates is shown. (H) Lung metastasis formation 14 days after intravenous injection B16F10 melanoma cells expressing pQa-dtSCT or vector control (B16-pQa-1 or B16-V). Dots over each bar represent individual mice, cumulative data from two independent experiments. Two-tailed unpaired t test was used (*=p < 0.05; **=p < 0.01).
DOI: https://doi.org/10.7554/eLife.38667.011

The following figure supplement is available for figure 6:

**Figure supplement 1.** Surface level of pQa-1 or Qa-1 on cells used in the assays in *Figure 6*.
DOI: https://doi.org/10.7554/eLife.38667.012

expression, which again could be blocked by the NKG2A/C/E-specific antibody (*Figure 6D and E*). This inhibitory effect could not be altered by the addition of Qdm peptide since Qdm is tethered to Qa-1 in the SCT format. Thus, pQa-1-SCT is functionally potent in inducing NKG2A-mediated NK cell inhibition which correlates well with its increased surface expression and thermostability (*Figure 5—figure supplement 5–1*). These findings are in agreement with the binding specificity of pQa-1 determined using pQa-1-dtSCT tetramer (*Figure 5*). To further assess whether inhibition of NKG2A$^+$ NK cells is mediated by pQa-1, CHO-V and CHO-pQa-1-SCT cells were treated with or without PI-PLC before co-culture with mouse spleen cells. We observed increased NKG2A$^+$ NK cell activation, as measured by IFNγ production and CD107a degranulation, when CHO-pQa-1-SCT was cleaved off the cell surface (*Figure 6F* and *Figure 6—figure supplement 6–1A*). Taken together, these data demonstrate that pQa-1 can functionally mimic Qa-1 to mediate inhibition of NK cells by specific engagement of the inhibitory receptor CD94/NKG2A.

## pQa-1 does not mimic antigen presentation to a Qa-1-restricted CD8$^+$ T cell clone

Recent reports suggest that HLA-E/Qa-1 can activate HLA-E/Qa-1 restricted CD8$^+$ T cells by presenting pathogen-derived or unconventional self-peptides (*Hansen et al., 2016*; *Bian et al., 2017*; *Oliveira et al., 2010*), which likely occurs when canonical MHC-I antigen presentation is impaired (*Oliveira et al., 2010*; *Nagarajan et al., 2012*; *Lampen et al., 2013*). Given this potential, we examined whether pQa-1 might 'unintentionally' activate Qa-1-restricted CD8$^+$ T cells. For this purpose, we used a well-characterized Qa-1-restricted CD8$^+$ T cell clone, Ln12 (*Doorduijn et al., 2018*). Co-culturing T2-Qa-1 cells, the TAP-deficient cell line transduced to express Qa-1 (*Figure 6—figure supplement 6–1B*) with Ln12 cells resulted in significant IFNγ production in a dose-dependent manner. However, under the same conditions, using T2 cells expressing pQa-1 (*Figure 6—figure supplement 6–1B*) we observed no T-cell activation (*Figure 6G*). Thus, it appears that in contrast to its mimicry of Qa-1 in CD94/NKG2A interactions, viral pQa-1 may not conserve antigen presentation properties with Qa-1.

## pQa-1 expression can protect cells from NK killing in vivo

The capacity of pQa-1 to provide protection in vivo was assessed in a murine melanoma metastasis model. B16F10 melanoma cells, which can establish metastasis in the lungs of C57BL/6 mice when NK control is subverted (*Takeda et al., 2011*), were transduced to express either empty vector alone (B16-V) or pQa-1-dtSCT (B16-pQa-1-dtSCT) (*Figure 6—figure supplement 6–1C*). The two resulting cell lines displayed no difference in proliferation in vitro. In vivo, B16F10 metastasis formation was prevented by NK cells as NK cell depletion with anti-NK1.1 resulted in high numbers of metastatic lesions 2 weeks after intravenous injection of the tumor cells (*Figure 6H*, compare B16-V to B16-V + anti-NK1.1). Consistent with the inhibitory effect of pQa-1 on NK cell activation in vitro, challenge of C57BL/6 mice with B16F10-pQa-1-dtSCT cells resulted in at least two-fold more lung metastases compared to mice challenged with B16F10 cells expressing empty vector (*Figure 6H*). These data thus demonstrate that pQa-1 has the capacity to protect cells from NK control in vivo.

## Downregulation of MHC-I by pK3 increases cell susceptibility to NK killing that can be subverted by pQa-1 co-expression

To experimentally assess if pK3-induced downregulation of MHC-I and Qa-1 can lead to NK activation and that pQa-1 functions to counteract these effects on NK cells, we transduced RMA cells with

either pK3 or the pK3 RING mutant. As expected, expression of pK3, but not pK3 RM induced dramatic reduction of surface classical MHC-I molecules as well as Qa-1 (*Figure 4B* and *Figure 7—figure supplement 7–1*). When cultured with syngeneic NK cells from mouse spleen, a significantly higher percentage of cells expressing pK3 were killed than cells expressing the pK3 RING mutant, demonstrating that a decrease of surface MHC-I induced by pK3 rendered cells susceptible to NK killing (*Figure 7A*). Moreover, in comparison to cells expressing only pK3, cells co-expressing pQa-1 with pK3 showed reduced susceptibility to NK cell killing. This reduction was not evident when NK cells from NKG2A$^{-/-}$ mice were used, indicating the protection provided by expression of pQa-1 is mediated by engagement of CD94/NKG2A (*Figure 7A*). Consistently, a small decrease in pQa-1 surface expression on target cells after PI-PLC treatment was found to correlate with a modest increase in NK killing (*Figure 7—figure supplement 7–2*). Together these data indicate that resistance of pQa-1 to pK3 is functionally important and suggest that RHVP has evolved a mechanism for thwarting 'missing self' recognition by NK cells. Collectively, our findings allow us to propose a model for how RHVP expresses two proteins that work in concert to evade CTL and NK killing (*Figure 7B*).

## Discussion

In this report, we identified and functionally characterized an RHVP encoded protein that can inhibit NK killing through direct engagement of CD94/NKG2A receptors. Although manipulation of host HLA-E/Qa-1 surface expression has been reported as a NK evasion strategy by other viruses, a viral mimic of Qa-1 has not been previously noted. The pQa-1 ORF is unusual in that the mRNA is produced by splicing of a primary transcript with an exon structure analogous to that found in mammalian MHC-I. The putative α1–3 domains of pQa-1 have >56% sequence identity to mouse and rat Qa-1, and it conserves the functional characteristics of Qa-1 including association with β2m and low level cell surface expression stabilized by Qdm or Qdm-like peptides. More impressively, pQa-1 mimics Qa-1 function by effectively engaging the inhibitory NK receptor CD94/NKG2A, yet is resistant to downregulation by the RHVP encoded MHC-I subversion protein pK3. These findings emphasize the collaboration between CTL and NK activation in control of RHVP infection and bring additional insight into the impact of host/pathogen interactions during coevolution.

The ability of herpesviruses to utilize RNA splicing as a means to expand their coding capacity was not appreciated until recently (*Stern-Ginossar et al., 2012*; *Conrad, 2009*). The development of next-generation sequencing techniques helped to uncover this viral strategy. For example, through mRNA-sequencing, up to 27 splicing junctions, corresponding to one or more introns in 17 viral genes (20% of genes) of KSHV were identified and seven of them were previously unrecognized (*Arias et al., 2014*). While the genome of RHVP was previously annotated to have three ORFs with sequence similarity to MHC-I (*Loh et al., 2011*), we now have shown that this region encoding pQa-1 has three consecutive exons separated by consensus splice donor and acceptor sites from which two short introns are removed. Our observation of spliced messenger RNA from RHVP is similar to what has been observed in KSHV, suggesting that the actual coding capacity and protein repertoire of γ−herpesviruses is likely larger and more elaborate than was previously appreciated.

The high degree of similarity in both sequence and function between pQa-1 and Qa-1 along with the conservation of the intron splicing sites suggest that the coding sequence of pQa-1 was acquired by horizontal transfer. This is clearly distinct from other known viral MHC-I-like proteins that have been previously identified in β−herpesviruses and implicated in CTL or NK evasion (*Revilleza et al., 2011*). Those viral MHC-I-like proteins, such as members of the m145 family of MCMV and UL18 of HCMV, possess a common MHC-I fold but are highly divergent in amino acid sequence (≤25% identity). In addition, their mRNA transcripts are not spliced, suggesting they may have been acquired from host transcripts as has been shown in other dsDNA viruses (*Grossegesse et al., 2017*; *Greijer et al., 2000*; *Bresnahan and Shenk, 2000*).

In contrast to Qa-1 and MHC-I proteins, pQa-1 lacks a canonical TM region and cytoplasmic tail. Its hydrophobic C terminus is removed during processing of the nascent protein with attachment of a GPI moiety, which is associated with resistance to the RHVP CTL evasion protein pK3 that appears to recognize classical and non-classical MHC-I by specific TM region interactions. This property of pQa-1 is physiologically significant. In this study, we demonstrate that expression of pK3 renders cells sensitive to NK killing that can be partly reversed by co-expression of pQa-1. A similar strategy exists in the host to counteract viral immune evasion proteins. For example, distinct from other

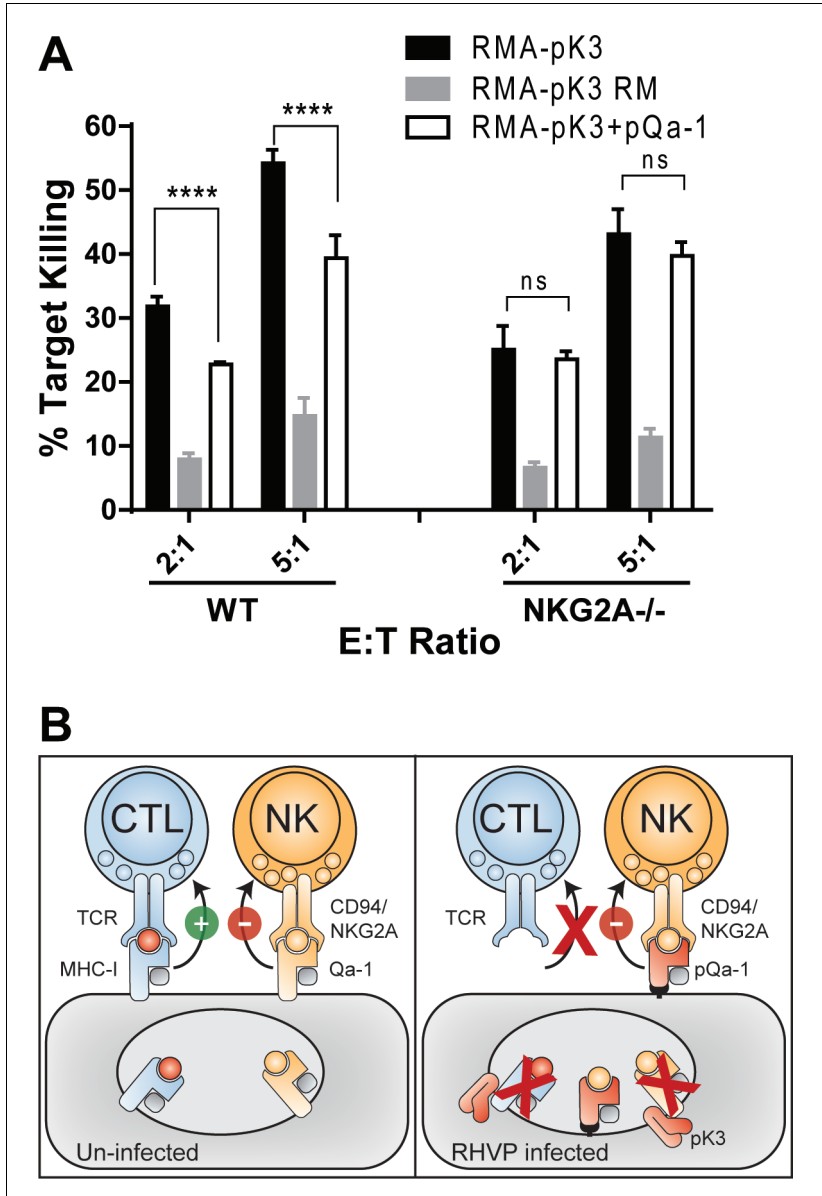

**Figure 7.** Downregulation of MHC-I by pK3 leads to NK killing susceptibility, while pQa-1 co-expression provides protection in an NKG2A-dependent manner. (**A**) The GFP$^+$ target cells (RMA cells expressing pK3 or pK3 RM by IRES-GFP vector or co-expressing pK3 and pQa-1) were mixed with the effectors (NK cells isolated from wildtype C57BL/6 or NKG2A$^{-/-}$ mice and activated by IL-2 for 6 days) at indicated E:T ratio and cultured at 37°C for 4 hr before stained with propidium iodide (PI) and analyzed by flow cytometry. NK killing%=[PI$^+$ target %/(PI$^+$ target% +PI$^-$ target%)*100]. Bars represent mean ±SD of four replicates. Dunnett's multiple comparisons test is used. (**B**) Working model for how RHVP-encoded pK3 and pQa-1 work in concert to evade CTL and NK killing. In normal circumstances, cytotoxic lymphocytes can survey foreign antigen through TCR recognition of MHC-I/peptide complexes at the cell surface, and NK cells can sense defects in antigen presentation and processing via engagement of HLA-E/Qa-1 by the inhibitory receptor CD94/NKG2A (left panel). In RHVP-infected cells, pK3 induces rapid degradation of MHC-I in the ER thus preventing CTL activation and clearance of infected cells; On the other hand, pQa-1 lacks an MHC-I like TM region and is thus resistant to pK3, which allows its cell surface expression and CD94/NKG2A engagement thus undermining 'missing self' recognition by NK cells (right panel).
DOI: https://doi.org/10.7554/eLife.38667.013

The following figure supplements are available for figure 7:

**Figure supplement 1.** Surface level of endogenous Qa-1, MHC-I and transduced pQa-1 on RMA cells expressing pK3 or pK3 RING mutant (pK3 RM).

*Figure 7 continued on next page*

Figure 7 continued

DOI: https://doi.org/10.7554/eLife.38667.014

**Figure supplement 2.** PI-PLC treatment diminishes pQa-1 mediated protection of RMA target cells expressing pK3.

DOI: https://doi.org/10.7554/eLife.38667.015

alleles that possess a TM and a cytoplasmic tail, MICA*008, the most frequently expressed allele of MICA in diverse populations worldwide, is GPI-anchored which is associated with its resistance to kK5, a MARCH ubiquitin E3 ligase similar to the pK3 encoded by KSHV. kK5-mediated downregulation of MICA requires lysine residues in the cytoplasmic tail of MICA for ubiquitination and subsequent internalization (*Thomas et al., 2008*; *Ashiru et al., 2009*). Thus, replacement of canonical TM and cytoplasmic tail with GPI anchor appears to be an adaptive modification to avoid harmful recognition events in both pathogens and hosts.

Compared to β−herpesviruses in which multiple proteins function to evade CTL and NK detection, most γ−herpesviruses in the genus rhadinovirus employ less complex strategies to avoid attack by CTL, and their NK evasion mechanisms are not well understood. A major CTL evasion mechanism in rhadinoviruses is carried out by a group of MARCH family ubiquitin ligases (*Hansen and Bouvier, 2009*). Notably, although they share similar domain organization and have MHC-I molecules as common targets for rapid degradation, the detailed mechanisms underlining how these viral MARCH ligases specifically recognize their substrates and where the degradation takes place are significantly different. For example, kK5 and kK3 of KSHV target mature MHC-I by conjugating a K63-linked ubiquitin chain on the tail of mature MHC-I heavy chain, which results in rapid endocytosis of MHC-I complex followed by its degradation in the lysosome (*Duncan et al., 2006*). In contrast, mK3 of γHV68 acts at an earlier point in MHC class I biosynthesis (*Boname and Stevenson, 2001*; *Yu et al., 2002*). It targets TAP-associated MHC-I in the ER by direct interaction with TAP, which results in K48-linked ubiquitination and subsequent degradation of MHC-I by the proteasome. While pK3 of RHVP also targets newly synthesized MHC-I for ER-associated degradation, it specifically recognizes the TM region of the MHC-I heavy chains. Functionally, pK3 not only induces dramatic loss of MHC-I but also TAP and tapasin (*Herr et al., 2012*).

As compared to the molecular mechanisms involved in MHC-I downregulation by the viral MARCH E3 ligases, KSHV, γHV68 and RHVP appear to have divergent strategies to avoid NK attack. For example, other than selectively targeting MHC-I to maintain expression of HLA-E, kK5 also downregulates MICA and MICB, the ligands of conserved activating NK receptor NKG2D, to evade NK cell cytotoxicity (*Thomas et al., 2008*). In addition, KSHV produces a miRNA that targets MICB at the transcriptional level (*Nachmani et al., 2009*). In contrast to kK3 and kK5, the target spectrum of mK3 is limited. It binds primarily TAP and only targets MHC-I heavy chains that are associated with TAP/tapasin complex (*Lybarger et al., 2003*). There have been no MHC-I-like proteins or NK inhibitory mechanisms identified in the genome of γHV68 thus far. In the case of RHVP, pK3 induces significant downregulation of classical MHC-I as well as Qa-1, yet it encodes a GPI anchored Qa-1-like protein that can suppress NKG2A[+] NK cells. Collectively these observations support the hypothesis that NK evasion strategies in γ−herpesviruses evolve with selection pressure imposed by CTL evasion and are intimately linked to the CTL subversion mechanisms employed by the virus.

In addition to its function in innate immunity, HLA-E/Qa-1 has been reported to play a role in adaptive immunity against intracellular pathogens or tumors by presenting microbial-derived peptides or self-peptides other than Qdm to HLA-E/Qa-1-restricted CD8[+] T cells (*van Hall et al., 2010*; *Joosten et al., 2016*; *Kraemer et al., 2014*). Whether pQa-1 can also be recognized by Qa-1-restricted CD8[+] T cells, which would be beneficial to the host but detrimental to the virus, may require further investigation in the context of viral infection. Nevertheless, our data show that pQa-1 expressed by a human TAP-deficient cell line does not activate a representative Qa-1-restricted T cell clone as does Qa-1. Further, the acidic alpha-3 domain CD loop (*Figure 1—figure supplement 1–2*) critical for interaction with CD8 (*Connolly et al., 1990*; *Chang et al., 2005*; *Wang et al., 2009*) and shared by classical MHC-I and Qa-1 is not conserved in pQa-1. Taken together it appears that in contrast to its mimicry of Qa-1 in CD94/NKG2A engagement, viral pQa-1 may not conserve the T-cell presentation features of cellular Qa-1. In summary, our findings reveal a novel NK evasion

mechanism in which a virus encodes a Qa-1-like protein capable of counteracting its CTL sabotage while retaining the ability to inhibit NK activity.

# Materials and methods

## Key resources table

| Reagent type (species) or resource | Designation | Source or reference | Identifiers | Additional information |
|---|---|---|---|---|
| Strain, strain background (*M. musculus*) | *Klrc1*[-/-] | PMID: 26680205 | | Dr. Marco Colonna laboratory (Washington University School of Medicine) |
| Strain, strain background (*M. musculus*) | C57BL/6NCr | Charles River Laboratories | Strain code: 556 | |
| Cell line (*M. musculus*) | B6/WT-3 | Dr. Stephen R Jennings (Louisiana State University Health Sciences Center); PMID: 6316651 | | an SV-40 transformed, C57BL/6 murine embryo fibroblasts line |
| Cell line (*M. musculus*) | BWZ.36 | Dr. Nilabh Shastri (University of California, Berkeley); PMID:8186188 | | a mouse T lymphoma line |
| Cell line (*M. musculus*) | FT1- (*Tap1*[-/-]) | Dr. Michael Edidin (Johns Hopkin University); PMID:10485658 | | a fibroblast line derived from *Tap1* knockout mice |
| Cell line (*M. musculus*) | 3KO (*H2-K*[-/-], *H2-D*[-/-], *B2m*[-/-] murine embryo fibroblast line) | Dr. Ted H. Hansen laboratory; PMID:12530981 | | 3KO mice were derived by breeding *H2-K*[b]/-*D*[b] doubly deficient mice (PMID:10229092) with *B2m*[-/-] mice (Jackson Laboratories) |
| Cell line (*Homo sapiens*) | T2 (174 x CEM.T2) | American Type Culture Collection | ATCC CRL-1992, RRID:CVCL_2211 | a TAP deficient T-B lymphoblast hybrid (PMID: 3522223) |
| Cell line (*M. musculus*) | RMA | PMID: 3877776 | RRID:CVCL_J385 | |
| Cell line (*M. musculus*) | B16-F10 | American Type Culture Collection | ATCC CRL-6475, RRID:CVCL_0159 | |
| Cell line (*Cricetulus griseus*) | CHO | Dr. Pamela Stanley laboratory (Albert Einstein College of Medicine) | | |
| Cell line (*Homo sapiens*) | 293F (FreeStyle 293 F Cells) | Thermo Fisher Scientific | R79007, RRID:CVCL_D603 | |
| Cell line (*Homo sapiens*) | 293T (HEK 293T) | American Type Culture Collection | ATCC CRL-3216, RRID:CVCL_0063 | |
| Antibody | FITC labeled anti-mouse NKG2A/C/E | eBioscience | cat# 11-5896-82 | clone: 20D5 (1:100) |
| Antibody | FITC labeled isotype control (RatIgG2a) | eBioscience | cat# 11-4321-85 | clone: eBR2a (1:100) |
| Antibody | eFluor 660 labeled anti-mouse CD107a | eBioscience | cat# 50-1071-82 | clone: eBio1D4B (1:1000 in culture) |
| Antibody | APC-eFluor labeled anti-mouse CD19 | eBioscience | cat# 47-0193-82 | clone: eBio 1D3 (1:100) |
| Antibody | PE-Cy7 labeled anti-mouse CD19 | eBioscience | cat# 25-0193-82 | clone: eBio1D3 (1:200) |

*Continued on next page*

*Continued*

| Reagent type (species) or resource | Designation | Source or reference | Identifiers | Additional information |
|---|---|---|---|---|
| Antibody | APC-eFluor 780 labled anti-mouse CD3e | eBioscience | cat# 47-0031-82 | clone: 145–2 C11 (1:100) |
| Antibody | Pacific Blue labeled anti-mouse CD3e | Biolegend | cat# 100214 | clone: 17A2 (1:100) |
| Antibody | PE-eFluor 610 labeled anti-mouse CD8a | eBioscience | cat# 61-0081-80 | clone: 53–6.7 (1:200) |
| Antibody | PE labeled anti-mouse CD94 | eBioscience | cat# 12-0941-82 | clone: 18D3 (1:100) |
| Antibody | mouse anti-HA tag | Covance | cat# MMS-101P | clone: 16B12 (1:500) |
| Antibody | eFluor 450 labeled anti-mouse IFNg | eBioscience | cat# 48-7311-82 | clone: XMG1.2 (1:100) |
| Antibody | PE-Cy7 labeled anti-mouse NK1.1 | eBioscience | cat# 25-5941-82 | clone: PK136 (1:100) |
| Antibody | PerCp Cy5.5 labeled anti-mouse NK1.1 | eBioscience | cat# 45-5941-82 | clone: PK136 (1:200) |
| Antibody | PE-labeled anti-mouse Thy1.1 | BD PharMingen | cat# 551401 | clone: OX-7 (1:150) |
| Antibody | biotin labeled anti-mouse Qa-1 | BD PharMingen | cat# 559829 | clone: 6A8.6F10.1A6 (1:200) |
| Recombinant DNA reagent | pMIG_pQa-1-HA | current study | | Schematic depiction is shown in *Figure 2A* |
| Recombinant DNA reagent | pMIN_pQa-1-HA | current study | | pMSCV.IRES.neo (pMIN) was described previously (PMID:15280476) |
| Recombinant DNA reagent | pMIG_pK3 | PMID:22403403 | | Dr. Ted H Hansen laboratory |
| Recombinant DNA reagent | pMIN_Qa-1 | current study | | DNA sequence encoding Qa-1b (NM_034528) was subcloned into pMIN vector |
| Recombinant DNA reagent | pMIG_pQa-1-SCT | current study | | Schematic depiction is shown in *Figure 5—figure supplement 1* |
| Recombinant DNA reagent | pFM1.2R_pQa-1-SCT-BirA | current study | | Described in the DAN constructs section in Materials and methods |
| Peptide, recombinant protein | pQa-1-SCT | current study | | Described in the Production of pQa-1-SCT Proteins and Tetramers section in Materials and methods |
| Commercial assay or kit | EasySep mouse NK cell isolation kit | StemCell Tecknologies | cat# 19815 | |

*Continued on next page*

*Continued*

| Reagent type (species) or resource | Designation | Source or reference | Identifiers | Additional information |
|---|---|---|---|---|
| Chemical compound, drug | PI-PLC (phosphatidylinositol-specific phospholipase C) | Sigma | P8804 | |
| Software, algorithm | PHYLIP Neighbor Joining algorithm | http://evolution.genetics.washington.edu/phylip/phylipweb.html | | |
| Others | rodent herpesvirus Peru (RHVP) | PMID: 21209105 | | Dr. Herbert W. Virgin laboratory in Washington University School of Medicine |

## DNA constructs

Two retroviral expression vectors, pMSCV.IRES.GFP (pMIG) and pMSCV.IRES.neo (pMIN) used for stable expression of pK3, Qa-1 and pQa-1-HA were described previously (*Lybarger et al., 2003*; *Wang et al., 2004*). The pQa-1 sequence was obtained from RHVP-infected MEF cells by RT-PCR. An HA tag was engineered in between the putative α3 domain and the last 26 amino acids (aa) (*Figure 2A*). The $L^d$-pQa-1 chimeric construct was made by fusing the sequence of H2-$L^d$ α1–3 to the C-terminal 26aa of pQa-1. The pQa-1-SCT construct consists of, beginning at the N terminus, the signal peptide of pQa-1, the Qdm peptide (AMAPRTLLL), linker1 (GGGASGGGGSGGGGS), the mature mouse β2m, linker2 ((GGGGS)$_4$) and the mature pQa-1-HA or the mature pQa-1 lacking the last 26aa followed by the BirA recognition peptide (GSTGLNDIFEAQKIEWHE). The HA-tagged pQa-1-SCT was expressed from the pMIG vector for surface expression in mammalian cells while the BirA-peptide-tagged version of pQa-1-SCT was produced from the pFM1.2R vector in 293F cells for biotin labeling (*Nelson et al., 2014*). All the constructs were confirmed by DNA sequencing.

## Cell lines and mice

Murine embryonic fibroblast B6/WT-3 (MEF), β2m-deficient cells ($B2m^{-/-}$), TAP1-deficient cells ($Tap1^{-/-}$, FT1-) and triple knockout fibroblasts ($H2-K^{b-/-}$ $H2-D^{b-/-}$ $β2m^{-/-}$; 3KO) were previously described (*Lybarger et al., 2003*) and of C57BL/6 ($H-2^b$) origin. Mouse RMA cells, B16F10, 293T and 293F were obtained from ATCC or Thermo Fisher Scientific. The BWZ.36 line was kindly provided by Dr. Nilabh Shastri (UC, Berkeley). CHO cell line is a gift from Dr. Pamela Stanley (Albert Einstein College of Medicine, NY). All cell lines have been assessed as mycoplasma negative. The pK3, pQa-1 and β2m were stably expressed in the indicated cells by pMIG or pMIN and were enriched or selected for GFP expression or neomycin resistance, respectively. 293F cells were cultured in Free-Style 293F expression medium (Gibco). All the other cells were cultured in complete RPMI1640 or DMEM (supplemented with 100 Unit/ml Penicillin–Streptomycin, 2 mM L-glutamine, 10 mM Hepes and 10% FBS) in a 5% $CO_2$ atmosphere at 37°C. C57BL/6 mice were purchased from Charles River laboratories, NKG2A$^{-/-}$ ($Klrc1^{-/-}$) mice (*Rapaport et al., 2015*) bred and housed in specific pathogen-free conditions in the accredited animal facilities at Washington University were a gift of Dr. Marco Colonna.

## Peptide-binding assay and peptides

MEF-pQa-1 cells were seeded in culture plates at $5 \times 10^5$ cells/ml. After overnight incubation at 37°C, peptides were added into the culture to a final concentration of 100 μM and incubated at 27°C, a condition known to promote peptide binding to classical MHC-I molecules (*Hansen and Myers, 2003*). Four hours later the cells were subjected to flow cytometric analysis. The OVA-derived peptide (SIINFEKL) (*Carbone and Bevan, 1989*), Qdm peptide (AMAPRTLLL), and others used for peptide binding (*Supplementary file 2*) were synthesized by GenScript (Piscataway, NJ).

## Production of pQa-1-SCT proteins and tetramers

Biotinylated soluble pQa-1-SCT or pQa-1-dtSCT monomer was produced by transient transfection of 293 F cells with the corresponding plasmid DNA as previously described (*Nelson et al., 2014*). The secreted protein containing a C-terminal 6His tag was purified from culture medium using Ni-agarose beads (Qiagen) followed by size-exclusion chromatography on a HiLoaD 26/60 Superdex 200 pg column (GE Healthcare). The proteins were biotin labeled on an included BirA-peptide-tag with enzyme in vitro following the manufacture's protocol (Avidity). The labeled protein was tested for biotinylation by ELISA using HRP conjugated streptavidin (Invitrogen), and then tetramerized through addition of phycoerythrin (PE)-labeled streptavidin (SA) (BD Biosciences) at a molar ratio of 4 molecules of SCT monomers to 1 molecule of PE-SA.

## Antibodies and flow cytometry analysis

Fluorescence-labeled antibodies including anti-NK1.1 (PK136), anti-NKG2A/C/E (20D5), anti-CD3 (145–2 C11), anti-CD19 (1D3), anti-CD8a (53–6.7), anti-CD107a (1D4B), anti-IFNγ (XMG1.2), anti-Thy-1.1 (OX-7), and isotype control were purchased from either eBioscience or BD Pharmingen. Unconjugated antibodies anti-H2-K$^b$ (B8-24-3) and anti-H2-L$^d$ (30-5-7) were described (*Lybarger et al., 2003*), anti-HA (16B12) and biotin-labeled anti-Qa-1 (6A8.6F10.1A6) were obtained from Covance and BD Pharmingen, respectively.

Spleen cells were first stained with fixable viability stain 660 (BD Horizon) or fixable viability dye eFluor 506 (eBioscience), followed by surface staining with directly conjugated antibodies diluted in Fc block (2.4G2) or with unconjugated/biotinylated antibodies in two steps using fluorescence conjugated secondary antibodies or streptavidin (BD Pharmingen), respectively. When intracellular staining was involved, the above surface staining was followed by fixation, permeabilization and directly conjugated antibody staining using Cytofix/Cytoperm kit (BD Biosciences) according to manufacturer's protocol. Stained samples were analyzed by flow cytometry using BD FACS Calibur, BD FACS Canto II or BD LSRFORTESSA X-20 and the data were analyzed with FlowJo software (Tree Star). The pQa-1-dtSCT tetramer staining was conducted at room temperature for 1 hr prior to the antibody staining. NK cells were defined as viable, NK1.1$^+$CD3$^-$CD19$^-$.

## NK stimulation assay

Co-cultures of $4 \times 10^6$ C57BL/6 splenocytes and $1 \times 10^5$ CHO cells expressing vector only, pQa-1, Qa-1, or pQa-1-SCT in the presence of 30 μM Qdm-k (AMVPRTLLL) peptide where indicated were incubated under 5% CO$_2$ atmosphere at 37°C. Where indicated 10 μg/ml isotype or 20D5 (anti-NKG2A/C/E) FITC-conjugated antibody was added to block the NKG2A-Qa-1 interaction. After 1 hr of 1x monensin (eBioscience) treatment in culture, 0.2 μg/ml fluorescent anti-CD107a antibody was added for an additional 5–8 hr. Cells were subsequently stained with fluorescent antibodies and analyzed by flow cytometry as described above.

## Experimental lung metastasis

A total of $3 \times 10^5$/mouse B16F10 cells transduced to express either empty vector or pQa-1-dtSCT were suspended in endotoxin-free PBS prior to i.v. injection into C57BL/6 mice. For NK-depletion controls, mice were administered i.p. with 200 μg anti-NK1.1 mAb, 2 days before and 2 days after B16F10 cell inoculation. After 14 days, the lungs were removed, PBS perfused and fixed with Feteke's solution (*Overwijk and Restifo, 2001*). The lung samples were blinded and the number of lung tumor metastatic colonies was counted under a dissecting microscope.

## NK cytotoxicity assay

NK cells were enriched from spleen cells of the indicated mice by negative selection using EasySep mouse NK Iso kit (StemCell Tech) according to manufacturer's instructions. After incubation in complete DMEM supplemented with 800 Units IL-2 and $2 \times 10^{-5}$ M 2-mercaptoethanol in 5% CO$_2$ atmosphere at 37°C for 6–9 days, the adherent cells (NK purity >95%) were collected and co-cultured with the indicated GFP$^+$ target cells at an E:T ratio of 2-6:1 for 4 hr. All the cells were then stained with propidium iodide (PI) 2–4 min before analysis by flow cytometry. Target killing%=[PI$^+$ target%/ (PI$^+$ target%+PI$^-$ target%)*100].

## Statistics

Statistical significance compared with a control group was calculated using ANOVA with Dunnett's multiple comparisons test, or unpaired t test, and annotated as $*=p < 0.05$; $**=p < 0.01$; $***=p < 0.001$; $****=p < 0.0001$.

## Acknowledgements

We thank Nilabh Shastri (UC Berkeley) for the BWZ.36 cell line, Marco Colonna (Washington University) for providing NKG2A$^{-/-}$ mice, and Alexander Barrow for helpful discussion. This work is supported by NIH grant R01-AI109687.

## Additional information

### Competing interests

Wayne M Yokoyama: Reviewing editor, *eLife*. The other authors declare that no competing interests exist.

### Funding

| Funder | Grant reference number | Author |
|---|---|---|
| National Institute of Allergy and Infectious Diseases | R01-AI109687 | Daved H Fremont |

The funders had no role in study design, data collection and interpretation, or the decision to submit the work for publication.

### Author contributions

Xiaoli Wang, Formal analysis, Supervision, Investigation, Visualization, Methodology, Writing—original draft; Sytse J Piersma, Formal analysis, Investigation, Visualization, Methodology, Writing—original draft; Christopher A Nelson, Conceptualization, Formal analysis, Visualization, Writing—review and editing; Ya-Nan Dai, Eric Lazear, Formal analysis, Investigation; Ted Christensen, Formal analysis, Investigation, Visualization; Liping Yang, Investigation, Methodology; Marjolein Sluijter, Formal analysis, Investigation, Methodology; Thorbald van Hall, Formal analysis, Investigation, Methodology, Writing—review and editing; Ted H Hansen, Wayne M Yokoyama, Conceptualization, Supervision, Funding acquisition, Writing—review and editing; Daved H Fremont, Conceptualization, Formal analysis, Supervision, Funding acquisition, Investigation, Writing—review and editing

### Author ORCIDs

Thorbald van Hall (iD) http://orcid.org/0000-0002-9115-558X
Wayne M Yokoyama (iD) http://orcid.org/0000-0002-0566-7264
Daved H Fremont (iD) http://orcid.org/0000-0002-8544-2689

### Ethics

Animal experimentation: This study was performed in strict accordance with the recommendations in the Guide for the Care and Use of Laboratory Animals of the National Institutes of Health. All of the animals were handled according to the approved institutional animal care and use committee (IACUC) protocol (#20130049). The protocol was approved by the Animal Studies Committee of Washington University.

### Decision letter and Author response

Decision letter https://doi.org/10.7554/eLife.38667.019
Author response https://doi.org/10.7554/eLife.38667.020

## Additional files

**Supplementary files**

• Supplementary file 1. RHVP encodes a protein with high identity to Qa-1.
DOI: https://doi.org/10.7554/eLife.38667.017

• Supplementary file 2. Peptides used for pQa-1 surface stabilization assay.
DOI: https://doi.org/10.7554/eLife.38667.016

### Data availability

All data generated or analysed during this study are included in the manuscript and supporting files.

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
