## [Decision Letter]

Thank you for submitting your article "Evasion of natural killer cell surveillance by a herpesvirus encoded Qa1-like protein" for consideration by *eLife*. Your article has been reviewed by three peer reviewers, including Michael L Dustin as the Reviewing Editor and Reviewer #1, and the evaluation has been overseen by Arup Chakraborty as the Senior Editor. The following individual involved in review of your submission has agreed to reveal their identity: Eric O Long (Reviewer #2).

The reviewers have discussed the reviews with one another and the Reviewing Editor has drafted this decision to help you prepare a revised submission.

Summary:

This is a substantial body of work that is, in general, well performed, with the main conclusion being that the pQa-1 is a viral inhibitor of NK cells that avoids down-regulation by viral proteins that target endogenous MHC class I and Qa-1.

Essential revisions:

A major issue that needs to be addressed is the title, which overreaches the results and should be modified to better fit the substantial findings of the paper. The key extrapolations are that pQa-1 results in evasion of surveillance, rather than inhibition of killing, and that the viral form of pQa1 was studied, rather than recombinant proteins derived from pQa1. The title is more of a working model that a succinct description of what is demonstrated. The experiments needed to test the model should be the subject of future papers, and you should correct the title in this instance.

If possible in a 2-month period, the author should add data showing expression for each pQa1 constructs tested, to demonstrate that PLC treatment results in increased NK cell killing or that pQa-1 doesn't protect cells that are enzymatically unable to make PGI anchored proteins and to address if pQa-1 needs endogenous proteins from TAP to be expressed. This is important because the herpes viruses also target TAP. Can pQa-1 be expressed in a TAP deficient cell? If these experiments are not at all possible it is felt that the work done could also be sufficient if the paper is very well edited to more precisely describe the data and the limitations of interpretation.

---

## [Author Response]

Essential revisions:A major issue that needs to be addressed is the title, which overreaches the results and should be modified to better fit the substantial findings of the paper. The key extrapolations are that pQa-1 results in evasion of surveillance, rather than inhibition of killing, and that the viral form of pQa1 was studied, rather than recombinant proteins derived from pQa1. The title is more of a working model that a succinct description of what is demonstrated. The experiments needed to test the model should be the subject of future papers, and you should correct the title in this instance.

As suggested, we have now changed the title of the paper from “Evasion of natural kill cell surveillance by a herpesvirus encoded Qa1-like protein” to "A herpesvirus encoded Qa-1 mimic inhibits natural killer cell cytotoxicity through CD94/NKG2A receptor engagement". We believe this new title accurately reflects the specific findings of our studies.

If possible in a 2-month period, the author should add data showing expression for each pQa1 constructs tested, to demonstrate that PLC treatment results in increased NK cell killing or that pQa-1 doesn't protect cells that are enzymatically unable to make PGI anchored proteins and to address if pQa-1 needs endogenous proteins from TAP to be expressed. This is important because the herpes viruses also target TAP. Can pQa-1 be expressed in a TAP deficient cell? If these experiments are not at all possible it is felt that the work done could also be sufficient if the paper is very well edited to more precisely describe the data and the limitations of interpretation.

We have added new data to our revised paper in Figure 6F that shows increased NKG2A^+^ NK cell activation (as measured by IFNγ production and CD107a degranulation) when CHO cells expressing pQa-1 SCT are treated with the GPI-cleaving enzyme PI-PLC. In support of these results, we have added an assessment of pQa-1 SCT surface expression by flow cytometry before and after PI-PLC treatment, which indicates significant removal of the protein from CHO cells (Figure 6—figure supplement 1A). Further, we also found that a small decrease in pQa-1 surface expression on RMA cells after PI-PLC treatment correlated with a modest increase in target killing in a syngeneic NK cytotoxicity assay (seen in new Figure 7—figure supplement 2). We note that the statistical significance of this experiment is small perhaps because the NK cell population used in the assay contains more than 50% NKG2A- cells that are unaffected by pQa-1. Further, during the 4- hour incubation of NK cells in the absence of PI-PLC, de novo synthesis of pQa-1 could potentially restore some pQa-1 surface expression thereby diminishing cytotoxicity.

As suggested, we also examined whether pQa-1 is expressed on the surface of TAP-deficient cells. We found, similar to what we had observed with wildtype MEF cells, that TAP-deficient MEFs stably transduced with pQa-1 express a low level on the cell surface, and this level can be enhanced by IFNγ treatment (data shown in new Figure 3D). We could also detect a significant level of pQa-1 on stably transduced T2 cells, a TAP-deficient human cell line (shown in new Figure 6—figure supplement 1B). Given the role peptide loading is known to play in MHC-I quality control, we would not expect pQa-1 to be efficiently released from the ER to the surface or to survive long at the cell surface without first being loaded with peptide. Thus, expression of pQa-1 on TAP-deficient cells suggests pQa-1 can load peptide independently of TAP as has been seen for Qa-1 and HLA-E.